# Spatial variation and attributable risk factors of anaemia among young children in Uganda: Evidence from a nationally representative survey

**Ronald Wasswa**[1,2]*, **Rornald Muhumuza Kananura**[1,2], **Hillary Muhanguzi**[3], **Peter Waiswa**[1,2]

**1** Department of Health Policy, Planning and Management, Makerere University School of Public Health, New Mulago Complex, Kampala, Uganda, **2** Centre of Excellence for Maternal and Newborn Health, Makerere University School of Public Health, New Mulago Complex, Kampala, Uganda, **3** Department of Local Government Statistics, Uganda Bureau of Statistics, Kampala, Uganda

* rwasswa93@yahoo.com

**Data Availability Statement:** All the data used in this study is available in a public, open access repository. These datasets are publicly available from the DHS website upon request (URL: https://

## Abstract

Anaemia continues to be a burden especially in developing countries that not only affects the physical growth and cognitive development of children but also increases their risk to death. Over the past decade, the prevalence of anaemia among Ugandan children has been unacceptably high. Despite this, spatial variation and attributable risk factors of anaemia are not well explored at national level. The study utilized the 2016 Uganda Demographic and Health Survey (UDHS) data with a weighted sample of 3805 children aged 6–59 months. Spatial analysis was performed using ArcGIS version 10.7 and SaTScan version 9.6. This was followed by a multilevel mixed-effects generalized linear model for the analysis of the risk factors. Estimates for population attributable risks (PAR) and fractions (PAF) were also provided using STATA version 17. In the results, intra-cluster correlation coefficient (ICC) indicates that 18% of the total variability of anaemia was due to communities within the different regions. Moran's index further confirmed this clustering (Global Moran's index = 0.17; *p-value<0.001*). The main hot spot areas of anaemia were Acholi, Teso, Busoga, West Nile, Lango and Karamoja sub-regions. Anaemia prevalence was highest among boy-child, the poor, mothers with no education as well as children who had fever. Results also showed that if all children were born to mothers with higher education or were staying in rich household, the prevalence would be reduced by 14% and 8% respectively. Also having no fever reduces anaemia by 8%. In conclusion, anaemia among young children is significantly clustered in the country with disparities noted across communities within different sub-regions. Policies targeting poverty alleviation, climate change or environment adaptation, food security as well interventions on malaria prevention will help to bridge a gap in the sub regional inequalities of anaemia prevalence.

www.dhsprogram.com/data/available-datasets.
cfm).

**Funding:** The authors received no specific funding
for this work.

**Competing interests:** The authors have declared
that no competing interests exist.

## Introduction

Anaemia in children under five years is defined as the low blood haemoglobin concentration
of less than 11g/dl [1]. It continues to be a public health concern because of its acute and long-
term adverse effect on the child's growth, development and survival [1, 2]. Globally, an esti-
mate of 273 million children have anaemia with over 10 million being in severe conditions [3].
In Africa, more than half of the children are anaemic (62%) [3]. This seemingly high propor-
tion ranks children in sub-Saharan Africa with the lowest mean blood haemoglobin concentra-
tion and therefore with the highest anaemia prevalence [3].

Uganda has one of the highest prevalence of anaemia among children aged 6–59 months
across the East African region [3], estimated at 53% in 2016 [4]. This rise follows an earlier
decrease in estimate from 73% to 49% in 2006 and 2011 respectively [4]. As a result, such anae-
mic children are subjected to stunted growth, impaired cognitive development, compromised
immunity, disability and increased risk of morbidity and mortality [2]. These high rates influ-
ence many children to live in discomfort situations which also underscore the high child health
burden in Uganda [2, 5–7].

Anaemia can be caused by iron deficiency, parasitic infections, acute and chronic infec-
tions, and other nutritional deficiencies (including folate, vitamin B12 and vitamin A) [1]. In
2018, at least 340 million children under 5 years were suffering from hidden hunger of vitamin
and mineral deficiencies [2]. It can also be acquired from inherited or acquired disorders that
affect haemoglobin synthesis, red blood cell production or red blood cell survival [1]. There-
fore, inability to access adequate food and care, low access to clean water, improper health ser-
vices, low sanitation, and repeated infections could influence a child's nutritional status as well
as the risk for anaemia [2].

Literature also shows that anaemia in children can be influenced by place of residence,
region, diarrhea or fever, low nutritional status of children, and household socioeconomic sta-
tus [8–11]. Maternal characteristics such as age, education level, number of births, religion,
access to media, nutritional indicators like underweight, and overweight have also been
reported to be important risk factors of anaemia [8, 10, 11]. These have been investigated
across different countries including sub-Saharan African region [9–12]. A study conducted in
India showed that children who had an experience of diarrhoea in seven days preceding the
survey were more likely to be anaemic [8]. Further, children born to uneducated mothers [8]
as well as those with fever or malaria have a higher risk of being anaemic [10]. Findings in
Ethiopia also show that the prevalence of anemia decreases as the age of children increases
with the fact that older children consume a variety of food sources rich in iron contents includ-
ing meats, poultry, fish, cereals [11]. In addition, children belonging to the highest and most
well-off households are associated with better nutritional status and are less vulnerable to get
anaemia [8]. It has been shown that children from poor households have both limited access to
diversified food and lower meal frequency than their counterparts in well-off households
which increases their risk to anaemia [9]. Spatial analysis and mapping of health indicators has
also been adopted [13–15]. This methodology has been noted as one of the effective tool for
programme managers and implementers in identifying communities that urgently require tar-
geted interventions as a way of meeting the sustainable development goals [13–15]. However,
much as the prevalence of anaemia is high, limited information is known on its spatial distri-
bution and the attributable risk factors so as to inform stronger recommendations in Uganda.

In this study, we therefore examine spatial variation and attributable risk factors of anaemia
among children aged 6–59 months in Uganda. Specifically, we sought to (a) determine and
evaluate spatial clusters of anaemia; (b) examine the sub-regional prevalence of anaemia and

the associated risk factors; (c) provide estimates of population attributable risk and population attributable fraction for each of the contributing factor.

## Materials and methods

### Data source and sampling strategy

Data used in this study is based from the 2016 Uganda Demographic and Health Surveys (DHS) that was downloaded from the DHS program website (https://dhsprogram.com/) after obtaining permission. The UDHS employs a multistage sampling design. The first stage involved random selection of clusters which acted as enumeration areas (EAs) that had been defined during the 2014 Population and Housing Census. The second stage involved a systematic selection of households from the clusters chosen during the first stage. In total, a national representative sample of 20880 households (30 per EA) from 696 clusters was randomly selected for the 2016 UDHS [4]. Sampling was also stratified to account for place of residence (rural vs urban households).

The 2016 UDHS collected information at both household and individual level covering a range of health indicators such as maternal and child health and nutrition [4]. Specifically, information was collected on children under 5 years and other members of the household including women aged 15–49, the structure of household dwelling, wealth index, feeding practices, vaccination, anthropometric measures of children, anemia testing, water supply and sanitation facilities. This study used the child dataset which contains information from household, and woman's questionnaires. In the 2016 UDHS, haemoglobin testing was performed on all children age 6–59 months in the sampled households using capillary blood. The testing of haemoglobin levels in the blood was done using a portable battery operated device called HemoCue Hb 201+ analyser with the consent taken from a parent or any guardian responsible for the child [4]. This study analyzed only children aged 6–59 months born at the time of the survey with accurate data on haemoglobin measurements. Also, children born to parents who were not permanent residents in the surveyed households were excluded since the study intended to also explore household characteristics. Fig 1 shows the final weighted sample size of the study.

### Study variables

**Outcome variable.** The dependent variable of the study is anaemia. Information on child anaemia was obtained by collecting blood specimens from all children aged 6–59 months for whom consent was obtained from their parents or the adult responsible. The variable of child anaemia was categorized into two with a code of 1 given to children whose haemoglobin level was less than 11 g/dl (considered anaemic) [1, 3] while the rest were coded 0 (considered not anaemic). This categorization was based on World Health Organization recommendation and threshold for anaemic children [1, 3].

**Exposure variables.** These include individual characteristics such as: maternal age, maternal education status, mother's working status, access to information, and pregnancy of the child wanted. Others were: child's age, sex of the child, size of child at birth, had diarrhea in the past 2 weeks, had fever in the past 2 weeks, ever been dewormed in the past 6 months, and ever taken vitamin A supplements in the past 6 months. The study also considered household factors (wealth index, number of children under five in a household, source of drinking water and type of toilet facility) as well as community variables (place of residence, sub-region, and distance to health facility).

During the 2016 UDHS, women were asked whether the pregnancy of the child at the time of conception was wanted then, wanted later or not wanted. This was the basis of generating

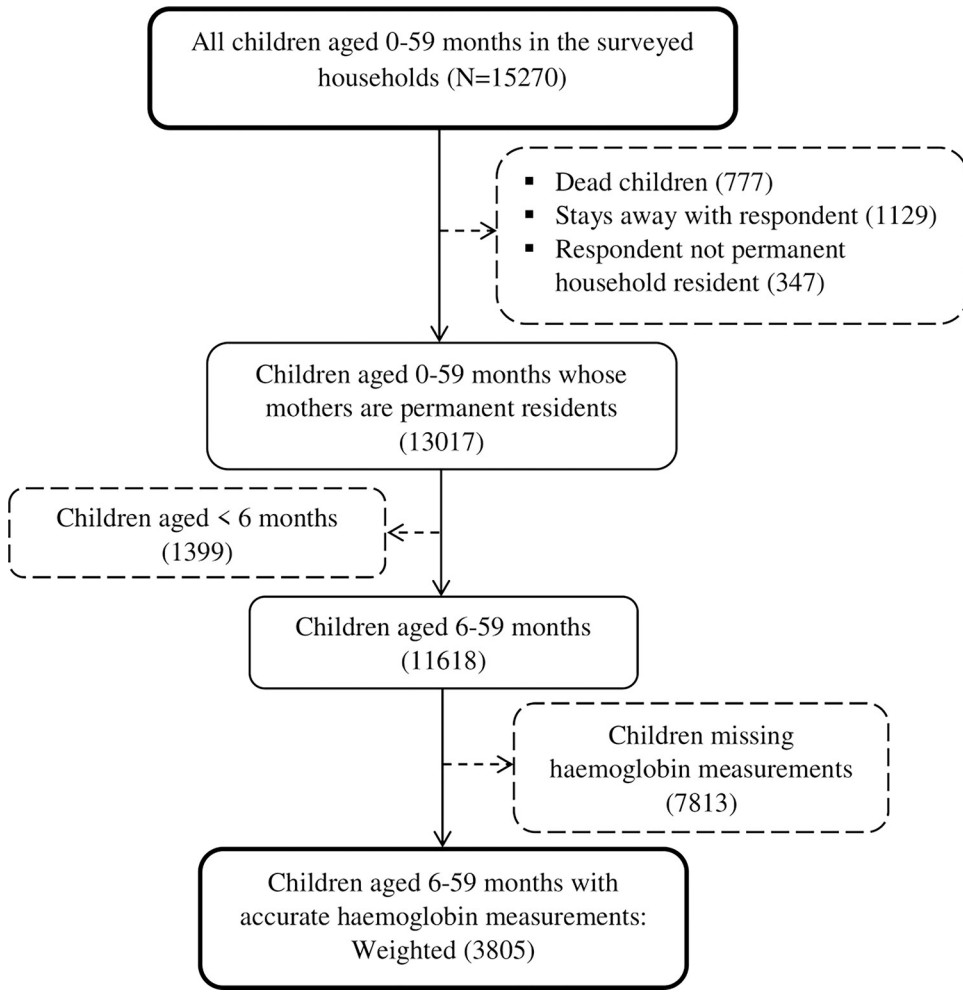

**Fig 1. Derivation of the sample size.**

the variable "pregnancy of the child wanted" with *yes* indicating that the pregnancy was wanted and *no* if otherwise. Mother's access to media was measured from the question that required women to state whether they had access to information on radio, television or newspapers. All responses to the forms of media were merged and coded 0 for "no" to all the sources, 1 for "yes" to any source of media. The selection of these variables was based on their significance in previous studies and their availability in the data used [8, 11, 16].

**Data analysis.** Data were first summarized descriptively using weighted frequencies, weighted percentages, mean and standard deviation. We generated a weighting variable using the sample weight variable in the DHS [4, 17]. The 'svy' command was also used to account for sampling weights, clustering, and stratification as required for complex survey data [17].

**Spatial analysis.** ArcGIS version 10.7 (ESRI, Redlands, CA, USA) and SaTScan version 9.6 statistical software were used for exploring the spatial distribution, global spatial autocorrelation, spatial interpolation, and for identifying significant hotspot areas of childhood anaemia. The weighted prevalence of child anaemia was computed for all the 696 clusters for the 2016 UDHS in STATA. These proportions were then imported and linked to the existing variables in the Uganda's geospatial location data (shapefile) using different software for analysis. The shapefile with sub-regional boundaries in Uganda was accessed and obtained from the DHS

program website. The DHS provides global positioning system (GPS) coordinates for clusters in order to facilitate spatial analyses while also maintaining participant confidentiality. These coordinates are displaced up to 2km in urban areas and 5km in rural areas, with up to 1% of points displaced up to 10km in rural areas [18]. The steps followed to produce maps have been described and documented elsewhere [13, 19].

**Spatial autocorrelation analysis.** The Global Moran's I statistic was used to measure the overall spatial autocorrelation of anaemia in the entire study area basing on the locations and attribute values of the data [20, 21]. Global Moran's I was calculated as [22];

$$I = \frac{n}{\sum_i^n \sum_j^n w_{ij}} \frac{\sum_i^n \sum_j^n w_{ij}(x_i - \bar{x})(x_j - \bar{x})}{\sum_i^n (x_i - \bar{x})^2} \tag{1}$$

Where $n$ is the number of spatial units; $x_i$ represents the value of the outcome variable $x$ at location $i$; $x_j$ represents the value of the outcome variable $x$ at another location $j$ (where $j \neq i$); $\bar{x}$ is the overall mean of the outcome variable in all locations; $w_{ij}$ represents the spatial weights matrix. This distance-based weight matrix is based on the inverse distance between locations $i$ and $j$ ($1/d_{ij}$). The value of the calculated Moran's I in Eq (1) ranges between -1 (perfect dispersion) and +1 (perfect correlation) with its interpretation based on its sign and magnitude. If the value is high positive and statistically significant ($p < 0.05$), then there is evidence of similar values (high or low) which are clustered also implying positive autocorrelation. On contrast, existence of high negative and significant Moran's I statistic implies dispersion and negative spatial autocorrelation (closely associated points are more dissimilar) [21]. A zero value of the Moran's I statistics indicates a random spatial pattern which is the null hypothesis in spatial dependence analysis. However, the global Moran's I is not sufficient enough to indicate areas within the map where specific types of values such as high or low are located or simply the nature of different clusters (or outliers).

**Cluster and outlier analysis.** We decomposed the global Moran's I in order to further identify local clusters and to examine their variation in relation to spatial autocorrelation across each of the study regions by computing the local Moran's index. This helps to examine the relationships between each observation and its surroundings as well as understand the regions or clusters that contribute more to the global spatial autocorrelation. The local Moran's Index, $I$, whose value also ranges between -1 and +1 for each unit $i$, was calculated as [23, 24].

$$I_i = \frac{x_i - \bar{x}}{\frac{\sum_{j=1}^n (x_j - \bar{x})^2}{n-1}} \sum_{j=1, j \neq i}^n w_{ij}(x_j - \bar{x}) \tag{2}$$

$$Z_i = \frac{I_i - E(I)}{\sqrt{V(I)}} \tag{3}$$

Where $Z_i$ is the z-score of Moran's I, E(I) is the expected value of I, V(I) is the variance of I, and $n$, $x_i$, $x_j$, $\bar{x}$ and $w_{ij}$ take up the same definitions in Eq (1).

This was followed by analysis of each of the computed localization with the values in the neighborhood using the Local Indicator of Spatial Association (LISA) [23]. In this approach, different clusters were identified, that is; clusters of high values (high-high), areas of low values (low-low) and the outliers. High–high areas are those with high value of cases and also surrounded by areas with high values while low–low refer to those with low value of cases and also surrounded by areas with low values of the outcome. Such areas were defined to have positive spatial autocorrelation. On contrary, all areas of low value cases and surrounded by those with high values (low–high), or vice-versa (high-low) were considered outliers and therefore

exhibit negative autocorrelation. A p-value <0.05 at 95% confidence interval was used as a significance level for this analysis.

**Hotspot analysis (Getis-Ord Gi\* statistic).** We then conducted hot spot analysis using Getis-Ord Gi\* statistics to show the variation as well identify cluster locations that have greater concentration of anaemia prevalence. This analysis generates z-scores and p-values that help to separate clusters of high values or high-risk areas (hotspot) from those of low values or low-risk areas (coldspot). The Getis-Ord Gi\* statistics is expressed as:

$$G_i^* = \frac{\sum_j^n w_{ij} x_j - \frac{\sum_j^n x_j}{n} \sum_j^n w_{ij}}{S \sqrt{\left(\frac{n \sum_j^n x_j^2}{n} - \left(\frac{\sum_j^n x_j}{n}\right)^2\right)\left(\frac{n \sum_j^n w_{ij}^2 - (\sum_j^n w_{ij})^2}{n-1}\right)}} \qquad (4)$$

Where $n$, $x_i$, $x_j$, $\bar{x}$ and $w_{ij}$ take up the same definitions in Eq (1). The statistical significance of this autocorrelation was determined by z-scores and p<0.05 with a 90%CI, 95% CI and 99% CI.

**Spatial scan statistics analysis.** The Bernoulli based model spatial scan statistical analysis was conducted to identify the most likely (primary) and secondary spatial clusters of child anaemia using Kulldorff's SaTScan version 9.6 software. All children who are anaemic were taken as cases while those who had no anaemia were considered as controls to fit the Bernoulli model. The SaTScan uses a circular scanning window that goes across the entire region of the study. For each window at any given position, the relative risk of anaemia among children inside was compared to that outside the window using the likelihood ratio test. P-values were generated using 999 Monte Carlo simulations to identify statistically significant clusters. This method has been widely adopted and described elsewhere [25].

**Spatial interpolation.** In order to understand the prevalence of anaemia in all areas of the country, we employed the ordinary kriging interpolation method to estimate the burden of anaemia in unsampled areas. This technique uses data in the sampled areas to predict the prevalence in the unsampled areas since it incorporates the spatial autocorrelation and it optimizes the weight statistically [19].

**Multilevel mixed effect analysis.** Due to the hierarchical nature of the DHS data, we employed a multilevel mixed-effects generalized linear model (using the logit link function) to examine the determinants of childhood anaemia. First, we ran a bivariate multilevel mixed-effects generalized linear model with the logit link function to analyze the bivariate associations. Crude odds ratios were presented and a p-value < 0.05 used to declare significant predictors. Only variables with p-value < 0.2 at this stage were fitted in the multivariable model.

At multivariate, four models were performed through a multilevel logistic regression with children at level 1 being nested within clusters (communities) at level 2. The communities are also nested within sub-regions. Model 1 (Empty model) was fitted without independent variables to test the random variability in the intercept and show the total variance in anaemia among children in different communities. Model II incorporated only individual/household characteristics while and model III separately had community variables. The last model (model IV) controlled for all individual/household and community variables level simultaneously. The results in the fixed part of each model were presented using odds ratios with their corresponding 95% confidence level. Similarly, the random part of the models was presented using variance estimate with their standard errors at 95% confidence level. Multicollinearity tests were performed to check the presence of correlations among explanatory factors using the variance inflation factor (VIF). The Intraclass Correlation Coefficient (ICC) was computed for each model to show the amount of clustering that are explained at each level of modeling.

Model comparisons were done using the Likelihood Ratio Test and Akaike information criteria (AIC). All the analyses were performed using STATA version 17 (StataCorp, College Station, Texas, United States).

**Population attributable risk and fraction.** The adjusted odds ratios for only statistically significant variables in the final model were used to estimate the population attributable fraction (PAF) and population attributable risk (PAR). This was computed using Miettinen's formula [26]:

$$PAF = p_r \left( \frac{OR_{adj} - 1}{OR_{adj}} \right) \qquad (5)$$

$$PAR = \frac{(OR_{adj} - 1)*p_e}{(OR_{adj} - 1)p_e + 1} \qquad (6)$$

Where $OR_{adj}$ is the adjusted odds ratio and $p_r$ is the prevalence of anaemia, $p_e$ is the proportion of anaemia cases exposed to the risk factor. These formula provide unbiased estimations of the population attributable risk in the presence of confounders [26] and have been adopted in similar health-related cross-sectional studies [27, 28].

The regpar and punaf commands in STATA were used to calculate the PAR and PAF respectively [29]. In addition, two scenario proportions: a baseline ("Scenario 0") and a fantasy ("Scenario 1") in which one or more exposure variables are presumed to be set to specific values are also estimated [29]. The PAR indicates the change in risk of anaemia within the overall population that would result from removal of the risk factor. On the other hand, PAF is interpreted as the proportion of anaemia that could be eliminated by removing or changing the distribution of these factors [26, 29, 30]. This helps to assess the relative importance of the factors as well as prioritize any possible intervention programmes.

**Ethical approval and consent to participate.** All data for this study is publicly available from the Demographic and Health Survey (DHS) website at www.measuredhs.com. We registered and requested for approval to access and download both main dataset and GPS data from the DHS website. Written informed consent from each interviewed mother was obtained during data collection. In addition, before extracting each of the child's blood samples that were used to perform the haemoglobin test, written informed consent were obtained from their mothers.

## Results

A total of 3805 under-five age children were considered in the study. Of these, one fifth (23.2%) were within the age range of 12–23 months with almost equal distribution of males (50.3%) and females (49.7%) as shown in Table 1. Almost 4 in 10 (36.5%) children were of birth order 1–2 while 20% had a small weight at birth. The results also show that half (53%) of the children were anaemic. Further, majority of the children were dewormed in the last 6 months (59.6%), had no fever in the last 2 weeks (61.2%) nor diarrhea (77.7%) and had been given vitamin A in the last 6 months (63.6%). Regarding maternal characteristics, almost half of the mothers were aged 25–34 years (46.8%), 76.5% had access to media, 3 in 5 (61.8%) had primary level of education and 8 in 10 (80.2%) were working. In the results, 4 in 10 (39.8%) of the pregnancies to these children were not wanted with 11% of the mothers having at least 3 births in the last 5 years before the survey. In terms of the household, at least 1 in 5 lie in the second or lowest wealth quantile, had three or more children aged under-five, had no improved water source and neither does any child under five sleep under a mosquito net.

**Table 1. Weighted distribution of the study variables.**

| Variable | Frequency (N = 3805) | Percentage (%) |
|---|---:|---:|
| *Child's characteristics* | | |
| **Child's age (in months)** | | |
| 6–11 | 478 | 12.6 |
| 12–23 | 883 | 23.2 |
| 24–35 | 855 | 22.5 |
| 36–47 | 815 | 21.4 |
| 48–59 | 774 | 20.4 |
| **Sex of the child** | | |
| Male | 1914 | 50.3 |
| Female | 1891 | 49.7 |
| **Child's birth order** | | |
| 1–2 | 1387 | 36.5 |
| 3–4 | 1104 | 29.0 |
| 5+ | 1314 | 34.5 |
| **Child's weight at birth** | | |
| Small | 761 | 20.0 |
| Average | 2017 | 53.0 |
| Large | 1028 | 27.0 |
| **Child dewormed (in the last 6 months)** | | |
| No | 1538 | 40.4 |
| Yes | 2267 | 59.6 |
| **Child had fever (in the last 2 weeks)** | | |
| No | 2327 | 61.2 |
| Yes | 1478 | 38.8 |
| **Child had diarrhea recently** | | |
| No | 2958 | 77.7 |
| Yes | 847 | 22.3 |
| **Child given vitamin A in the last 6 months** | | |
| No | 1387 | 36.4 |
| Yes | 2418 | 63.6 |
| **Child anaemic** | | |
| No | 1776 | 46.7 |
| Yes | 2029 | 53.3 |
| *Maternal factors* | | |
| **Mother's age** | | |
| 15–24 | 1112 | 29.2 |
| 25–34 | 1780 | 46.8 |
| 35–49 | 913 | 24.0 |
| **Mother's education level** | | |
| No education | 430 | 11.3 |
| Primary | 2353 | 61.8 |
| Secondary | 784 | 20.6 |
| Higher | 237 | 6.3 |
| **Mother currently working** | | |
| No | 754 | 19.8 |
| Yes | 3051 | 80.2 |
| **Pregnancy of child wanted** | | |

(*Continued*)

**Table 1.** (Continued)

| Variable | Frequency (N = 3805) | Percentage (%) |
|---|---|---|
| Yes | 2290 | 60.2 |
| No | 1515 | 39.8 |
| **Mother's access to media** | | |
| No access | 895 | 23.5 |
| Has access | 2910 | 76.5 |
| **Number of births in the last 5 years** | | |
| 1 | 1471 | 38.7 |
| 2 | 1902 | 50.0 |
| 3+ | 432 | 11.3 |
| *Household characteristics* | | |
| **Household wealth index** | | |
| Lowest | 861 | 22.6 |
| Second | 786 | 20.7 |
| Middle | 744 | 19.6 |
| Fourth | 670 | 17.6 |
| Highest | 743 | 19.5 |
| **Children under 5 in a household** | | |
| 1 | 1124 | 29.5 |
| 2 | 1851 | 48.7 |
| 3+ | 829 | 21.8 |
| **Water source** | | |
| Improved | 2950 | 77.5 |
| Not improved | 855 | 22.5 |
| **Toilet facility** | | |
| Improved | 1349 | 35.5 |
| Not improved | 2455 | 64.5 |
| **Children under 5 slept in a mosquito net** | | |
| All | 2244 | 59.0 |
| Some | 482 | 12.7 |
| No | 1079 | 28.4 |
| *Community factors* | | |
| **Distance to the health facility** | | |
| Big problem | 1586 | 41.7 |
| Not a big problem | 2219 | 58.3 |
| **Place of residence** | | |
| Urban | 798 | 21.0 |
| Rural | 3006 | 79.0 |
| **Sub-region** | | |
| Kampala | 117 | 3.1 |
| South Buganda | 480 | 12.6 |
| North Buganda | 397 | 10.4 |
| Busoga | 366 | 9.6 |
| Bukedi | 268 | 7.0 |
| Bugisu | 176 | 4.6 |
| Teso | 234 | 6.2 |
| Karamoja | 98 | 2.6 |
| Lango | 227 | 6.0 |

*(Continued)*

**Table 1.** (Continued)

| Variable | Frequency (N = 3805) | Percentage (%) |
|---|---|---|
| Acholi | 190 | 5.0 |
| West Nile | 266 | 7.0 |
| Bunyoro | 225 | 5.9 |
| Tooro | 349 | 9.2 |
| Ankole | 295 | 7.7 |
| Kigezi | 118 | 3.1 |

Relatedly, majority of the households (64.5%) do not have an improved toilet facility. Concerning community factors, 41.7% of the mothers find a big problem to access a health facility. Also, about 8 in 10 (79.0%) of the children were in rural areas with about 12.6% in South Buganda whereas only 2.6% and 3.1% were from Karamoja and Kampala respectively (Table 1).

## Random effect and spatial analysis of anaemia

Results in Table 2 show that the variation in anaemia among children is heterogeneous at both at regional and community level. The intra-cluster correlation coefficient indicates that 18% of the total variability of anaemia is due to communities within regions. Results from Moran's index also confirm that anaemia among children was generally clustered.

## Hotspots and cold spots of anaemia

Fig 2 presents the hot spots and cold spots of anaemia among children aged 6–59 months. In the results, children in Acholi, Teso, Busoga, West Nile, Lango and Karamoja sub-regions were at a higher risk of having anaemia (hotspot). Further, Ankole, Kigezi, Tooro, Bugisu and some parts of Bunyoro were cold spots (low risk) areas of anaemia among children (Fig 2). Also to note, the Acholi and Busoga had the highest number of communities with anaemia followed by West Nile and Lango (S1 Table). On the other hand, the communities with low anaemia outcome were in Ankole, Kigezi and Bugisu.

## Spatial interpolation of anaemia

Based on the Kriging interpolation technique, Acholi, Karamoja, West Nile, Teso, most parts of South Buganda and Busoga predicted higher rates of childhood anaemia. However, Ankole, Kigezi, Bugisu, Bunyoro and some parts of North had lower predicted rates of anaemia (Fig 3).

## Spatial scan statistics

Further, a total of two statistically significant (one primary/most likely and the other secondary) scanning windows were identified in the spatial scan analysis to be significant (p<0.05) (Fig 4). The primary cluster spatial window with a total sampled population of 1446 covered West Nile, Acholi, Lango, Teso, Karamoja, and parts of Bugisu, Busoga, and Bunyoro. A list of affected districts in each of the sub-regions has been presented in the (S2 Table). In total, this spatial window consists of 226 communities with a log-likelihood ratio (LLR) of 43.1, relative risk (RR) of 1.3 at p< 0.01. It showed that children within this window were 1.3 times more likely to have a higher risk of anaemia compared to children outside the area. The secondary scanning window was found at the border of Busoga and North Buganda containing 4

**Table 2. Random effect and spatial analysis of anaemia among children.**

| Random effect from Model 1 | |
|---|---|
| Regional level variance (SE) | 0.20(0.09) |
| Region > Community level variance (SE) | 0.52(0.09) |
| Regional ICC(%) | 5.1 |
| Community \| Region ICC(%) | 18.0 |
| **Spatial autocorrelation of anaemia** | |
| Moran's Index | 0.171 |
| Z score | 8.345 |
| p-value | 0.000 |

ICC is the intra-cluster correlation coefficient; SE is the standard error; random effect results are based on an empty model where children are nested in communities and communities nested in regions.

communities with RR of 1.9, LLR = 13.6. This showed that the risk of anaemia among children within this window was twice higher than those outside (S2 Table).

## Risk factors of anaemia among young children

Table 3 presents the multivariate multilevel analysis of risk factors of anaemia among children. We present these findings basing on model 3 that adjusted for all individual, household and community factors because of its low AIC in comparison to the rest. The model presents a total variance of communities within regions of 0.54 as well as an intra-class correlation of

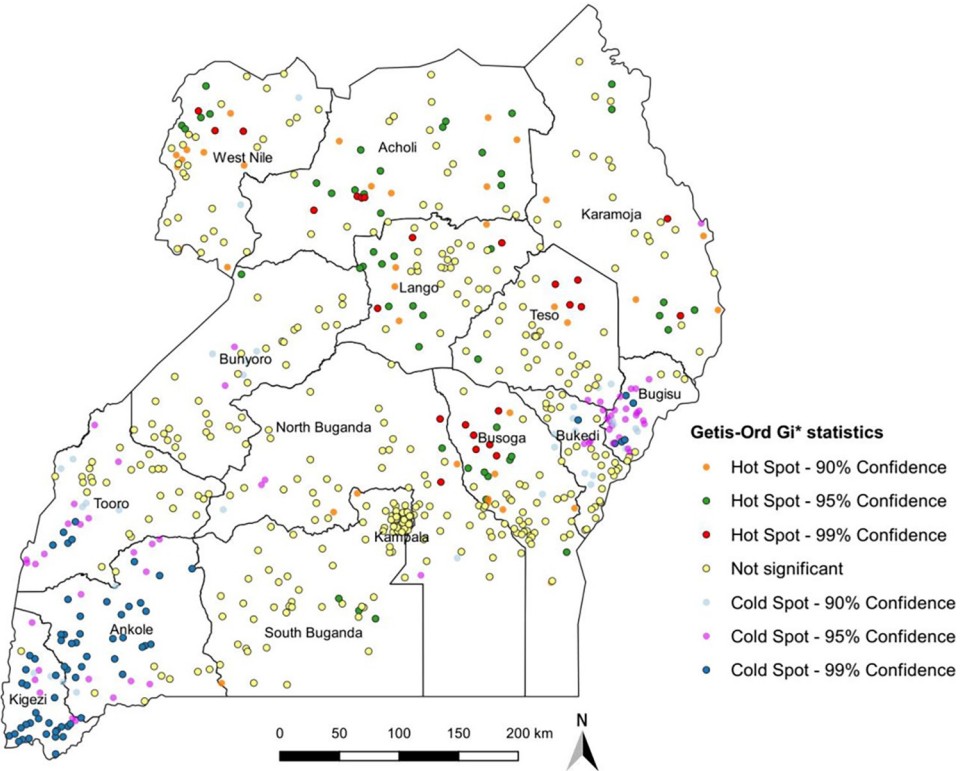

**Fig 2. Hot spots and cold spots of childhood anaemia, 2016 UDHS: All shapefiles requested and obtained from DHS program website (https://dhsprogram.com/).**

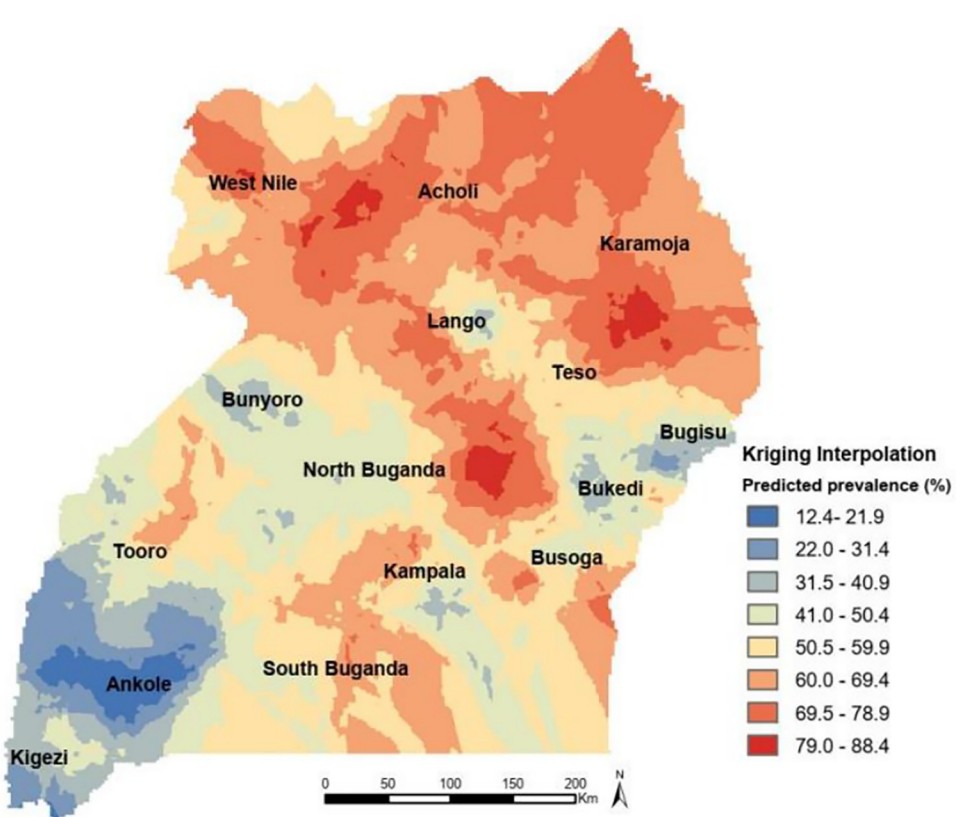

**Fig 3. Ordinary kriging interpolation of anaemia among young children, 2016 UDHS: All shapefiles requested and obtained from DHS program website ([https://dhsprogram.com/](https://dhsprogram.com/)).**

16%. Child's age, sex, being dewormed, having fever, mother's education level and household wealth index were important correlates of anaemia among children aged 6–59 months at a 5% level of significance.

Specifically, children in the age groups 24–35, or 36–47 and 48–59 months were less likely to be anaemic compared to their counterparts aged 6–11 months (AOR = 0.28, 95%CI = 0.17–0.47; AOR = 0.22, 95%CI = 0.15–0.34; AOR = 0.19, 95%CI = 0.12–0.31) respectively. Female children (AOR = 0.86, 95%CI = 0.78–0.95) and those who were dewormed (AOR = 0.83, 95% CI = 0.69–0.99) were less likely to anaemic. Results also revealed that having had fever in the 2 weeks increases the odds of child anaemia (AOR = 1.56, 95%CI = 1.27–1.91). Compared to mothers with no education, children whose mothers had primary or higher education had reduced odds of anaemia (AOR = 0.69, 95%CI = 0.50–0.96; AOR = 0.50, 95%CI = 0.37–0.67) respectively. The study also shows that as household wealth improves, the risk of child anaemia reduces. Children in the second (AOR = 0.66, 95%CI = 0.52–0.84), middle (AOR = 0.56, 95% CI = 0.42–0.75), fourth (AOR = 0.52, 95%CI = 0.40–0.69) or highest (AOR = 0.50, 95% CI = 0.37–0.67) wealth quintile were less likely to be anaemic compared to those in the lowest quintile.

## Population attributable risk and fraction of anaemia

The PAF and PAR were calculated by considering scenario 1 (a hypothetical scenario in which all children were in the most advantageous position. Results indicate a reduction in anaemia prevalence among the oldest children, the female, those dewormed, children with no fever, as

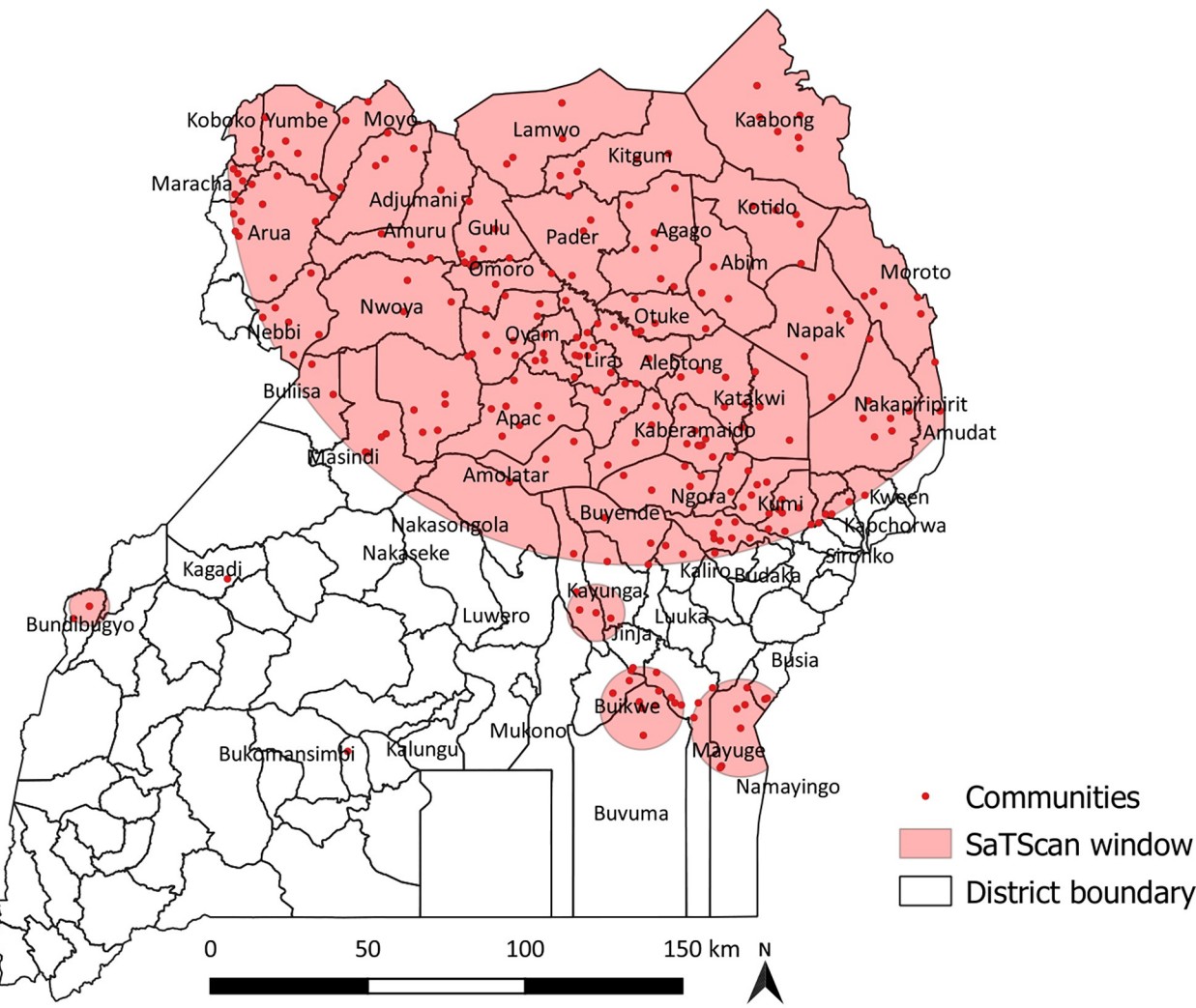

**Fig 4. SaTScan analysis anaemia among young children, 2016 UDHS: All shapefiles requested and obtained from DHS program website (https://dhsprogram.com/).**

well as those whose mothers have higher education or are staying in better-off households. Therefore, if all children were aged 48–59 months, were female, had been dewormed, or had no fever in the last two weeks, anaemia prevalence would be reduced by 24%, 3%, 3%, and 8% respectively. Further, if all children under study were born to mothers with higher education or were staying in the richest households, the anaemia prevalence would reduce by 14% and 8% respectively (Table 4).

## Discussion

This study examined the spatial variation and risk factors of anaemia among children aged 6–59 months. It further provides estimates for the attributable risks and preventable measures. In the results, the prevalence of anaemia was 53% similar to that reported in the 2016 UDHS [4]. The study also confirmed that anaemia was spatially clustered and affected by child, maternal, household and community factors.

The spatial scan statistics identified 226 most likely primary communities mainly in West Nile, Acholi, Lango, Teso, and Karamoja sub-regions. There were also 4 secondary clusters

**Table 3. Association between anaemia among children and the selected variables.**

| Variable | Model 2 aOR(95%CI) | Model 3 aOR(95%CI) | Model 4 aOR(95%CI) |
|---|---|---|---|
| **Child's age** | | | |
| 6–11[†] | 1.00 | | 1.00 |
| 12–23 | 0.79(0.55–1.13) | | 0.79(0.55–1.13) |
| 24–35 | 0.28(0.17–0.47)*** | | 0.28(0.17–0.47)*** |
| 36–47 | 0.23(0.15–0.34)*** | | 0.22(0.15–0.34)*** |
| 48–59 | 0.19(0.12–0.31)*** | | 0.19(0.12–0.31)*** |
| **Sex of the child** | | | |
| Male[†] | 1.00 | | 1.00 |
| Female | 0.86(0.78–0.95)*** | | 0.86(0.78–0.95)** |
| **Child dewormed in the last 6 months** | | | |
| No[†] | 1.00 | | 1.00 |
| Yes | 0.83(0.69–0.99)* | | 0.83(0.69–0.99)* |
| **Child had fever in the last 2 weeks** | | | |
| No[†] | 1.00 | | 1.00 |
| Yes | 1.56(1.27–1.91)*** | | 1.56(1.27–1.91)*** |
| **Child had diarrhea recently** | | | |
| No[†] | 1.00 | | 1.00 |
| Yes | 1.01(0.81–1.24) | | 1.01(0.82–1.24) |
| **Mother's age** | | | |
| 15–24[†] | 1.00 | | 1.00 |
| 25–34 | 0.82(0.63–1.07) | | 0.82(0.63–1.06) |
| 35–49 | 0.72(0.46–1.12) | | 0.72(0.46–1.12) |
| **Mother's education level** | | | |
| No education[†] | 1.00 | | 1.00 |
| Primary | 0.69(0.49–0.95)* | | 0.69(0.50–0.96)* |
| Secondary | 0.68(0.44–1.07) | | 0.69(0.43–1.09) |
| Higher | 0.50(0.37–0.67)** | | 0.50(0.37–0.67)*** |
| **Mother currently working** | | | |
| No[†] | 1.00 | | 1.00 |
| Yes | 0.79(0.56–1.10) | | 0.79(0.57–1.10) |
| **Pregnancy of child wanted** | | | |
| Yes[†] | 1.00 | | 1.00 |
| No | 1.06(0.94–1.20) | | 1.05(0.93–1.19) |
| **Mother's access to media** | | | |
| No access[†] | 1.00 | | 1.00 |
| Has access | 0.98(0.68–1.42) | | 0.99(0.68–1.42) |
| **Number of births in the last 5 years** | | | |
| 1[†] | 1.00 | | 1.00 |
| 2 | 1.34(0.96–1.87) | | 1.34(0.96–1.86) |
| 3+ | 1.19(0.53–1.80) | | 1.18(0.78–1.79) |
| **Household wealth index** | | | |
| Lowest[†] | 1.00 | | 1.00 |
| Second | 0.66(0.53–0.84)*** | | 0.66(0.52–0.84)** |
| Middle | 0.56(0.42–0.75)*** | | 0.56(0.42–0.75)*** |
| Fourth | 0.52(0.39–0.68)*** | | 0.52(0.40–0.69)*** |
| Highest | 0.48(0.37–0.64)*** | | 0.50(0.37–0.67)*** |
| **Children under 5 in a household** | | | |

(*Continued*)

**Table 3.** (Continued)

| Variable | Model 2 aOR(95%CI) | Model 3 aOR(95%CI) | Model 4 aOR(95%CI) |
|---|---|---|---|
| 1[†] | 1.00 | | 1.00 |
| 2 | 0.96(0.67–1.37) | | 0.96(0.68–1.37) |
| 3+ | 1.21(0.92–1.58) | | 1.21(0.92–1.58) |
| **Water source** | | | |
| Improved[†] | 1.00 | | 1.00 |
| Not improved | 0.81(0.68–0.98)* | | 0.81(0.67–1.07) |
| **Toilet facility** | | | |
| Improved[†] | 1.00 | | 1.00 |
| Not improved | 0.99(0.78–1.27) | | 0.99(0.78–1.26) |
| **Children under 5 slept in a mosquito net** | | | |
| All[†] | 1.00 | | 1.00 |
| Some | 1.28(0.82–2.00) | | 1.28(0.82–2.01) |
| No | 1.17(0.92–1.50) | | 1.17(0.92–1.50) |
| **Distance to the health facility** | | | |
| Big problem[†] | | 1.00 | 1.00 |
| Not a big problem | | 0.84(0.73–0.98)* | 0.89 (0.75–1.06) |
| **Place of residence** | | | |
| Urban[†] | | 1.00 | 1.00 |
| Rural | | 1.24(1.05–1.47)* | 1.02(0.83–1.25) |
| **Random effect** | | | |
| Regional level variance (SE) | 0.10(0.06) | 0.19(0.09) | 0.10(0.06) |
| Region > Community level variance (SE) | 0.54(0.10) | 0.51(0.09) | 0.54(0.10) |
| Regional ICC(%) | 2.7 | 4.7 | 2.6 |
| Community | Region ICC(%) | 16.4 | 17.5 | 16.3 |
| Model fit statistics | | | |
| Log likelihood | -2283 | -2515 | -2282 |
| AIC | 4593 | 5040 | 4591 |

[†] is a Reference category; aOR is the adjusted odds ratio; SE is the standard error; ICC is the inter-cluster correlation coefficient; AIC is the Akaike Information Criterion; 95% CI is the Confidence Interval; *$p<0.05$, **$p<0.01$, *** $p<0.001$; the assessment was based on multivariate three-level logistic regression model

with children most likely to be anaemic in Busoga and North Buganda. The kriging interpolation further predicted the prevalence of anaemia to be highest in the north and eastern part of the country. This is due to differences in the multidimensional poverty levels. Northern region is the poorest part of the country with Karamoja leading followed by Acholi and West Nile sub-regions [31, 32]. As of 2020, an estimated 85%, 64%, 59%, and 57% of children in Karamoja, Acholi, West Nile, and Lango live in households below the national poverty line [32]. This region is also arid/semi-arid and therefore food crop growing is a challenge. Next to the north in deprivation, is the eastern part of the country. The study showed that Busoga region has the highest risk of anaemia within the east. Not only being poor, land in the region is mainly used for cultivation of sugarcane and food crops growing is on a very small scale [33]. In addition, income generated from cultivation is inadequate for the population to be able to purchase food or to meet its needs. This scenario is totally different from Ankole or Kigezi regions which are food-baskets of the country. This therefore could explain the regional differences of anaemia in this study. Thus, policies targeting poverty alleviation, climate change adaptation and food security (including integration of cash crop growing into food production) for sustainable livelihood should be implemented.

**Table 4. Estimated population attributable risk and population attributable fraction of anaemia among children aged 6–59 months.**

| Variables | Scenario 1%(95%CI) | PAR% | PAF% |
|---|---|---|---|
| **Child's age** | | | |
| **6–11**[†] | | | |
| **12–23** | 69.4(66.0–72.8) | -16.0 | -30.1 |
| **24–35** | 47.7(44.2–51.6) | 5.6 | 10.5 |
| **36–47** | 43.3(39.7–47.4) | 1.0 | 18.7 |
| **48–59** | 40.5(36.9–44.4) | 12.9 | 24.1 |
| **Sex of the child** | | | |
| Male[†] | | | |
| **Female** | 51.6(49.3, 54.0) | 1.7 | 3.1 |
| **Child dewormed in the last 6 months** | | | |
| **No** | | | |
| **Yes** | 51.6(49.4, 53.8) | 1.7 | 3.2 |
| **Child had fever in the last 2 weeks** | | | |
| Yes[†] | | | |
| No | 49.3(47.1, 51.5) | 4.0 | 7.5 |
| **Mother's education level** | | | |
| No education[†] | | | |
| Primary | 52.8(50.6, 54.9) | 1.0 | 1.0 |
| Secondary | 53.3(49.1, 57.4) | 0.0 | 0.0 |
| Higher | 46.1(38.0, 54.3) | 7.3 | 13.6 |
| **Household wealth index** | | | |
| Lowest[†] | | | |
| Second | 54.6(50.8, 58.4) | -1.3 | -2.5 |
| Middle | 49.2(45.3, 53.1) | 4.1 | 7.7 |
| Fourth | 48.6(44.6, 52.5) | 4.8 | 8.9 |
| Highest | 48.9(43.2, 54.7) | 4.4 | 8.2 |

The baseline prevalence (Scenario 0) = 53.3%; Scenario 1 is the prevalence of the exposure variables at each of the fantasy category; PAR is the population attributable risk; PAF is the population attributable fraction. Analysis was controlled for maternal, child, household and community characteristics.

In the multilevel analysis, children aged more than 23 months were at lower odds of being anaemic. The trend of anaemia prevalence reduced for each age group of children aged 24–35, 36–47, and 48–59 months. This finding is consistent with many previous studies in Ethiopia [12]. This could be explained by the fact that as children get older, they are progressively exposed to more quality or richer and complete diet with sufficient intake of iron. This helps in the prevention of iron-deficiency or anaemia.

Our findings on associations between the sex of the child and anaemia is aligned with previous studies [34, 35]. The possible reason for this require further investigation. Besides, programs attempting to intervene nutritional problems among the vulnerable population should be reconsidered.

Consistent with other studies [36–38], children who were dewormed were less likely to be anaemic. Worms may compete for nutrients in the body including vitamin A and as a result induce chronic intestinal blood loss [36, 37]. Therefore, deworming prevents any infections that cause of iron loss which prevents anemia.

The study also found that children who had fever were at an increased risk of being anaemic. This result was in agreement with previous studies [10–12]. This could be attributed to the malaria and other parasitic infections. Moreover in Uganda, almost half (49%) of households either have no insecticide treated nets or do not have enough for all household members [4]. Secondly, 4 in 10 children under five do not sleep under a mosquito net and yet this is one of the preventive measures of malaria [4].

Like other previous studies [8, 16], maternal education was found to be negatively associated with childhood anaemia. Different studies have found that maternal education is key in reducing anaemia among children. The possible reason for this finding could be that educated mothers may be knowledgeable on various essential nutrition behaviors including feeding habits for their children that may be important in preventing anaemia. Also, educated mothers have better health-seeking behavior for childhood illnesses and can have better work opportunities for consequently better income as compared to uneducated mothers. These help the child to have a better environment as well as improved quality diet.

Consistent with other studies[8, 10–12, 16], children living in households of higher wealth quintile were less likely to be anaemic. A study by Sharma and colleagues in India revealed socio-economic inequality as a key player in the increased anaemia with poor children suffering a disproportionate burden [8]. Children in poor households may not be able to receive iron rich foods like animal foods and vitamin-rich foods especially vitamins A and C which are very important for iron absorption [12].

Regarding the attributable risk estimates, being dewormed, having no fever, higher maternal education, and highest wealth quintile showed the largest contributors in the reduction of anaemia prevalence. Similar factors have also been reported as key attributes to other health outcomes elsewhere [27, 28, 39].

## Conclusion

The findings of the study provide comprehensive evidence on the spatial variation of anaemia prevalence and the associated risk factors among children aged 6–59 months in Uganda. It further provides estimates for the attributable risks and preventable measures. Child's age, sex, deworming, fever, mother's education level, and household wealth index were the key risk factors. Policies on sensitization and those targeting poverty alleviation, climate change or environment adaptation, food security as well interventions on malaria prevention will help to bridge a gap in the sub regional inequalities of anaemia prevalence.

## Strength and limitation

The potential strength of our study is the use of a nationally representative sample of all children aged 6–59 months in Uganda making the results generalizable. Secondly, the different methodologies adopted to analyze the spatial distribution and attributable risk as well as a multilevel analysis due to the nested nature of the data provided reliable estimates. However, there were some limitations to this study. The cross-sectional nature of the data could not permit the establishment of causality. Further, since some of the information in the DHS is self-reported, there could be tendency of recall bias especially with issues related to study variables like age, episodes of fever in children as well as deworming and uptake of vitamin A supplements. There could also exist some measurement errors and misclassification because the locations of DHS clusters are randomly displaced to protect the confidentiality of survey respondents. In spite of these limitations, the authors believe that the findings from this study will enhance proper public health interventions that can reduce the prevalence of anaemia in the affected areas.

## Supporting information

**S1 Table. Hot spot and cold spot analysis of anaemia among communities per sub-region.** (DOCX)

**S2 Table. SaTScan analysis result of hot spot areas of anaemia among young children in Uganda.** (DOCX)

## Acknowledgments

We express our gratitude the demographic and health survey program for allowing us to access the data set for this study.

## Author Contributions

**Conceptualization:** Ronald Wasswa, Rornald Muhumuza Kananura, Hillary Muhanguzi, Peter Waiswa.

**Data curation:** Ronald Wasswa, Rornald Muhumuza Kananura, Hillary Muhanguzi, Peter Waiswa.

**Formal analysis:** Ronald Wasswa, Rornald Muhumuza Kananura, Peter Waiswa.

**Investigation:** Ronald Wasswa, Rornald Muhumuza Kananura, Hillary Muhanguzi, Peter Waiswa.

**Methodology:** Ronald Wasswa, Rornald Muhumuza Kananura, Peter Waiswa.

**Software:** Ronald Wasswa, Rornald Muhumuza Kananura.

**Supervision:** Rornald Muhumuza Kananura, Peter Waiswa.

**Validation:** Ronald Wasswa, Hillary Muhanguzi, Peter Waiswa.

**Visualization:** Ronald Wasswa, Rornald Muhumuza Kananura, Peter Waiswa.

**Writing – original draft:** Ronald Wasswa, Rornald Muhumuza Kananura, Hillary Muhanguzi, Peter Waiswa.

**Writing – review & editing:** Ronald Wasswa, Rornald Muhumuza Kananura, Hillary Muhanguzi, Peter Waiswa.

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
