## [Decision Letter · Decision Letter 0]

7 Feb 2023

PGPH-D-22-01906

Spatial variation and preventable risk factors of anaemia among young children in Uganda: evidence from a nationally representative survey

Dear Dr. Wasswa,

Thank you for submitting your manuscript to PLOS Global Public Health. After careful consideration, we feel that it has merit but does not fully meet PLOS Global Public Health’s publication criteria as it currently stands. Therefore, we invite you to submit a revised version of the manuscript that addresses the points raised during the review process.

The design and statistical analyses are valid, and the manuscript is well-written. The reviewers have requested minor revisions to be done before the manuscript is accepted for publications. I look forward to your revised work.

We look forward to receiving your revised manuscript.

Kind regards,

Paddy Ssentongo, MD, PhD, MPH

Academic Editor

Journal Requirements:

1. In the online submission form, you indicated that "All the data used in this study is available in a public and open access repository. This dataset is publicly available from the DHS website upon request (URL: https://www.dhsprogram.com/data/available-datasets.cfm) on request.". All PLOS journals now require all data underlying the findings described in their manuscript to be freely available to other researchers, either 1. In a public repository, 2. Within the manuscript itself, or 3. Uploaded as supplementary information.

2. Some material included in your submission may be copyrighted. According to PLOS’s copyright policy, authors who use figures or other material (e.g., graphics, clipart, maps) from another author or copyright holder must demonstrate or obtain permission to publish this material under the Creative Commons Attribution 4.0 International (CC BY 4.0) License used by PLOS journals. Please closely review the details of PLOS’s copyright requirements here: PLOS Licenses and Copyright. If you need to request permissions from a copyright holder, you may use PLOS's Copyright Content Permission form.

Potential Copyright Issues:

Figures2,4,5 and 6: please (a) provide a direct link to the base layer of the map (i.e., the country or region border shape) and ensure this is also included in the figure legend; and (b) provide a link to the terms of use / license information for the base layer image or shapefile. We cannot publish proprietary or copyrighted maps (e.g. Google Maps, Mapquest) and the terms of use for your map base layer must be compatible with our CC-BY 4.0 license. 

Additional Editor Comments (if provided):

Reviewers' comments:

Reviewer's Responses to Questions

**Comments to the Author**

1. Does this manuscript meet PLOS Global Public Health’s publication criteria? Is the manuscript technically sound, and do the data support the conclusions? The manuscript must describe methodologically and ethically rigorous research with conclusions that are appropriately drawn based on the data presented.

Reviewer #1: Yes

Reviewer #2: Yes

Reviewer #3: Yes

2. Has the statistical analysis been performed appropriately and rigorously?

Reviewer #1: Yes

Reviewer #2: Yes

Reviewer #3: Yes

3. Have the authors made all data underlying the findings in their manuscript fully available (please refer to the Data Availability Statement at the start of the manuscript PDF file)?

Reviewer #1: Yes

Reviewer #2: Yes

Reviewer #3: Yes

4. Is the manuscript presented in an intelligible fashion and written in standard English?

Reviewer #1: Yes

Reviewer #2: Yes

Reviewer #3: Yes

5. Review Comments to the Author

Reviewer #1: The authors conducted a cross-sectional study utilizing the 2016 Uganda Demographic and Health Survey (UDHS) data to evaluate the spatial variation and preventable risk factors of anaemia among children aged 6-59 moths in Uganda. The findings of the study provide comprehensive evidence on the spatial variation of anaemia prevalence and the associated risk factors among children aged 6-59 months in Uganda.The authors provide a well-written paper with thoughtful analysis and discussion. Regarding the research strategy and findings, reviewers have several comments that should be addressed prior to publication.

Abstract

The results and conclusions revolve around the spatial variation and risk factors of anaemia in Uganda, however, the background rationale doesn't touch on these a priori relationships. It would be helpful to revise the abstract background with a concept focusing on the research gap (the 3rd-5th paragraph of the Introduction)..

Introduction

1. The prevalence and factors associated with anaemia in sub-Saharan African countries have been largely investigated, the rationale for study purpose would been benefited from the extended discussion of these observations (ref 8-12).

Methods

A few additional pieces of information included in the methods would help contextualize the results.

1) Is the sample of 20,880 households national representative? Please further explain.

2) The citations from UDHS or previous UDHS papers would be helpful to introduce the data source and sampling strategy.

3) It would be helpful to list all investigated exposure at the beginning of the paragraph (Exposure variables).

4) The extended description for how sample was weighted would be beneficial for data analysis.

Results

Both the number and the percentage are expected to be descripted in the text.

Discussion

It is worthwhile to further discuss why primary clusters mainly in West Nile and other regions, and proposing the possible public health recommendations regarding the observation to decrease the prevalence of anaemia in these areas is expected (2nd paragraph and 9th paragraph in Discussion).

Tables and Figures

1. The unit (months or years) of age is expected to add in Table 1.

2. There are a significant number of Figures. It is suggested to put some of them as supplemental materials.

Reviewer #2: Reviewer’ s Report

“Spatial variation and preventable risk factors of anaemia among young children in Uganda: evidence from a nationally representative survey” is an excellent piece of research manuscript in field of child health of Africa. I don’t have any further queries regarding the objective, methods and analysis of this research as these are chosen very carefully. There is no problem in language and coherence as well from my perspective. Therefore, I suggest to accept this manuscript for the publication. However, the authors should answer the following question before the acceptance of manuscript.

What is the reason to choose the Cluster and outlier analysis and Hotspot analysis and even SaTScan Analysis? If all these analyses are necessary discuss on the results obtained from these methods? I also expect to have some discussion on Spatial interpolation. I also suggest similar analysis from other regions of the world. For example:

https://www.mdpi.com/1660-4601/19/14/8664

Reviewer #3: Thank you for the opportunity to review this manuscript. The authors aimed to investigate the spatial variation and preventable risk factors of anaemia among children aged 6-59 months in Uganda. The manuscript is very well written and easy to follow. The objectives were clearly met using appropriate and thorough statistical methods. The authors are commended for their efforts. I recommend minor revision based on a few very minor grammatical errors in the manuscript. In addition, the authors should specify more clearly the random effect incorporated into the GLMM. Based on the discussion of the results, it appears to be based on the cluster. However, when describing the multilevel mixed effect model, this is not made clear.

6. PLOS authors have the option to publish the peer review history of their article (what does this mean?). If published, this will include your full peer review and any attached files.

**Do you want your identity to be public for this peer review?** For information about this choice, including consent withdrawal, please see our Privacy Policy.

Reviewer #1: No

Reviewer #2: No

Reviewer #3: **Yes: **Danielle Jade Roberts

---

## [Editor Report · Decision Letter 1]

19 Apr 2023

Spatial variation and attributable risk factors of anaemia among young children in Uganda: evidence from a nationally representative survey

PGPH-D-22-01906R1

Dear Mr Wasswa,

We are pleased to inform you that your manuscript 'Spatial variation and attributable risk factors of anaemia among young children in Uganda: evidence from a nationally representative survey' has been provisionally accepted for publication in PLOS Global Public Health.

Best regards,

Paddy Ssentongo, MD, PhD, MPH

Academic Editor

I have accepted the manuscript, however please make the following minor revisions before publications.

1) Please update the abstract with the revised version

2) Replace "poor" and "rich" with "lowest" and "highest" household wealth index quintile